# An Improved Segmentation Method for Automatic Mapping of Cone Karst from Remote Sensing Data Based on DeepLab V3+ Model

**Han Fu [1,2], Bihong Fu [1,*] and Pilong Shi [1]**

[1] Key Laboratory of Digital Earth Science, Aerospace Information Research Institute, Chinese Academy of Sciences, Beijing 100094, China; fuhan2017@radi.ac.cn (H.F.); shipl@aircas.ac.cn (P.S.)

[2] University of Chinese Academy of Sciences, Beijing 100049, China

* Correspondence: fubh@aircas.ac.cn; Tel.: +86-10-8217-8096

**Abstract:** The South China Karst, a United Nations Educational, Scientific and Cultural Organization (UNESCO) natural heritage site, is one of the world's most spectacular examples of humid tropical to subtropical karst landscapes. The Libo cone karst in the southern Guizhou Province is considered as the world reference site for these types of karst, forming a distinctive and beautiful landscape. Geomorphic information and spatial distribution of cone karst is essential for conservation and management for Libo heritage site. In this study, a deep learning (DL) method based on DeepLab V3+ network was proposed to document the cone karst landscape in Libo by multi-source data, including optical remote sensing images and digital elevation model (DEM) data. The training samples were generated by using Landsat remote sensing images and their combination with satellite derived DEM data. Each group of training dataset contains 898 samples. The input module of DeepLab V3+ network was improved to accept four-channel input data, i.e., combination of Landsat RGB images and DEM data. Our results suggest that the mean intersection over union (MIoU) using the four-channel data as training samples by a new DL-based pixel-level image segmentation approach is the highest, which can reach 95.5%. The proposed method can accomplish automatic extraction of cone karst landscape by self-learning of deep neural network, and therefore it can also provide a powerful and automatic tool for documenting other type of geological landscapes worldwide.

**Keywords:** UNESCO natural heritage site; cone karst landscape; segmentation; deep learning; multi-source remote sensing data

## 1. Introduction

Karst landscapes are general term for the surface and underground landforms formed by the dissolution of water on soluble rocks. They are widely distributed in the world and have a total area of 1.25 million km$^2$ in China [1]. Mapping of a karst landscape generally aims to record its location, distribution, type and development stage in an area [2]. The formation of karst landscape is influenced by various factors, acting together or alone, such as local lithology, geological structure, climate, precipitation and vegetation. Most of karst areas have complex topography and geomorphic conditions, fragile ecological environment and poor accessibility, which significantly restrict the development of regional land use, urban planning, mineral survey, geological disaster prevention and control, ecological environment protection and tourism resources management [3]. Therefore, extracting the geomorphic information and spatial distribution of karst landscapes effectively is essential and significant for land use planning, ecological environment protection, and management for these heritage sites.

Currently, extraction of karst landscapes is traditionally conducted through field investigations or interpretation using remote sensing images and digital elevation model (DEM) data. Based on methods of visual interpretation [4,5], supervised classification [6],

ratio calculation [7], decision tree and maximum likelihood estimation (MLE) [8] applied to satellite remote sensing images, one can extract information of rocky desertification in karst areas. By using DEM data, researchers can extract various terrain factors, such as rivers [9,10], elevation [9,10], slope [4,11], aspect [4,11], for karst landform recognition. These traditional methods not only consume a lot of time, cost and human resources, but also their classification results of karst landscapes depend on experts' knowledge and the accuracy is approximately 80% [12]. Both optical remote sensing images and DEM data were not jointly used for mapping karst landscapes [13,14].

Recently, deep learning (DL) has emerged as the state-of-the-art machine learning technique with a great capability of remote sensing image classification. Instead of depending on manually-engineered shallow features, deep learning techniques automatically learn hierarchical features (from low-level to high-level) from input data directly [15,16]. However, due to the lack of training data, semantic segmentation based on deep learning is usually used to recognize features with regular shape or distinct boundaries, such as buildings [17], urban green space [18] and shorelines [19]. It is a big challenge for recognition of complex landscapes. The karst landscape, as one typical complex landscape, the geometry and texture information contained in remote sensing data and elevation information contained in DEM data are all important factors for segmentation [13].

DeepLab model is a convolutional network for image semantic segmentation proposed by Google in 2015 based on fully convolutional networks (FCNs) [20]. So far, it has four versions, i.e., DeepLab V1 [21], DeepLab V2 [22], DeepLab V3 [23] and DeepLab V3+ [24]. The three main advantages of the DeepLab system are efficiency, accuracy and simplicity, by mixing various new algorithms, such as conditional random fields (CRF) [25–27], atrous convolution [28], atrous spatial pyramid pooling (ASPP) [29,30], spatial pyramid pooling (SPP) [31] and encode-decoder structure, to the deep convolutional neural networks (DCNNs). The latest version of DeepLab V3+ released in 2018 has a mean intersection over union (MIoU) of 89% in the PASCAL visual object classes (VOC) [32] dataset. In this study, we selected the DeepLab V3+ as the basic network model of training. However, a limitation of applying the DeepLab model is it only allows to input one-channel or three-channel data as training samples. If we need to use optical remote sensing images containing red, green, and blue channels information and additionally the DEM data, there will be four channels information, which is not accepted by the currently used DeepLab V3+ model. In order to extract information of karst landscapes using jointly remote sensing and DEM data for higher accuracy, the input layer of DeepLab V3+ model needs to be improved so that the network could accept at least four-channel data.

Therefore, in this study we attempt to develop an automatic method to extract spatial distribution of cone karst landscape by combining satellite remote sensing images and DEM data based on the deep learning model DeepLab V3+ in order to improve landscape classification accuracy and efficiency. The South China Karst, with an area of approximately 550,000 km$^2$, as the world's most spectacular example of humid tropical to sub-tropical karst landscape, was named as a United Nations Educational, Scientific and Cultural Organization (UNESCO) world natural heritage property in 2007 and 2014, respectively. It includes seven karst clusters, namely, Shilin, Libo, Wulong, Guilin, Shibing, Jinfoshan, and Huanjiang Karsts, which contain the most typical types of karst landforms, including tower karst, pinnacle karst, cone karst and so on. We applied the developed method to classify cone karst in Libo, which is a representative of tropical cone karst landscape in the world.

Following the Introduction, detailed description of the Libo study area is presented in Section 2. In Section 3, the developed methodology to extract cone karst landscape using both satellite remote sensing images and DEM data based on the DeepLab V3+ model is described. Then, the segmentation results of cone karst landscape in Libo using the developed method are evaluated. To further highlight the advantage of using the DL method to classify cone karst landscape, we also conducted the same experiment but using the classical classification method, i.e., Support Vector Machine (SVM) and compared with

the results achieved using the DeepLab V3+ model. In Section 5, we briefly discussed the reason that combing both satellite remote sensing images with DEM data can significantly improve accuracy of segmentation of cone karst landscape. The conclusions are drawn in the last section.

## 2. Study Area

The Libo cone karst landscape, as one of South China Karst UNESCO heritage sites in 2007, is located in the southernmost part of Guizhou Province, with an area of 2431.8 km$^2$ (107°37′~108°39′E, 25°06′~25°39′N). Due to influence of tectonic deformation during the Cenozoic era, the terrain displays higher in west and lower in east, with an average elevation of 747 m above sea level (m a.s.l). The topographic information of Libo area derived from the Advanced Spaceborne Thermal Emission and Reflection Radiometer Global Digital Elevation Model (ASTER GDEM) data is shown in Figure 1a. Figure 1b shows the geographical location of study area.

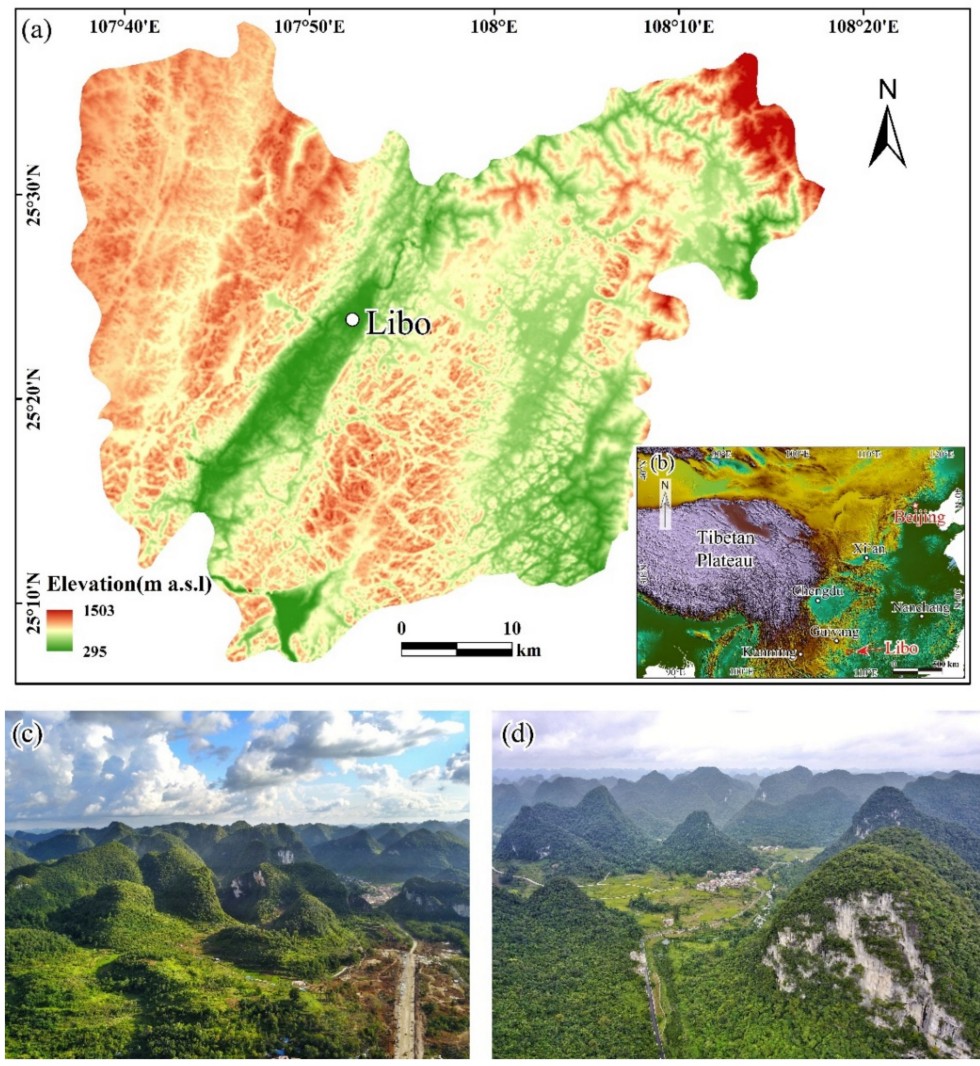

**Figure 1.** (**a**) ASTER GDEM image showing the topographic features of Libo area. (**b**) An index map showing the geographical location of study area. (**c**,**d**) Two images acquired by Unmanned Aerial Vehicle (UAV) over the typical cone karst landscape in Libo heritage site.

It represents the world's most spectacular cone karst landscape, which satisfies the evaluation standards of Articles VII and VIII of the World Heritage and has extraordinary aesthetic value and scientific research values [33,34].

As product of erosion and collapse of carbonate rocks by water, karst landscapes have different surface morphology at different developmental stages. The cone karst landscape in Libo heritage site consists of cone-shaped peaks connected by pedestals and depressions, dolines or canyons between cones, with an average slope of 45° and different heights [35], as shown in Figure 1c,d.

Tectonically, Libo cone karst landscape belongs to the slotted fold belt in the southern Yangtze Block. The folds are mainly extending NNE direction. Geological interpretations show that they are composed of gentle synclines and narrow anticlines as shown in Figure 2, which are typical wide-spaced folds. Particularly, the NWW and NNE-striking conjugate joints and faults are also developed well in the Libo area (Figure 2). These folds, faults and joints controlled the spatial distribution of the drainage system in the study area, and the main river flows are mostly parallel with the structural trend, forming the unique landform along the NNE-striking wide-spaced fold belts in Libo [36,37].

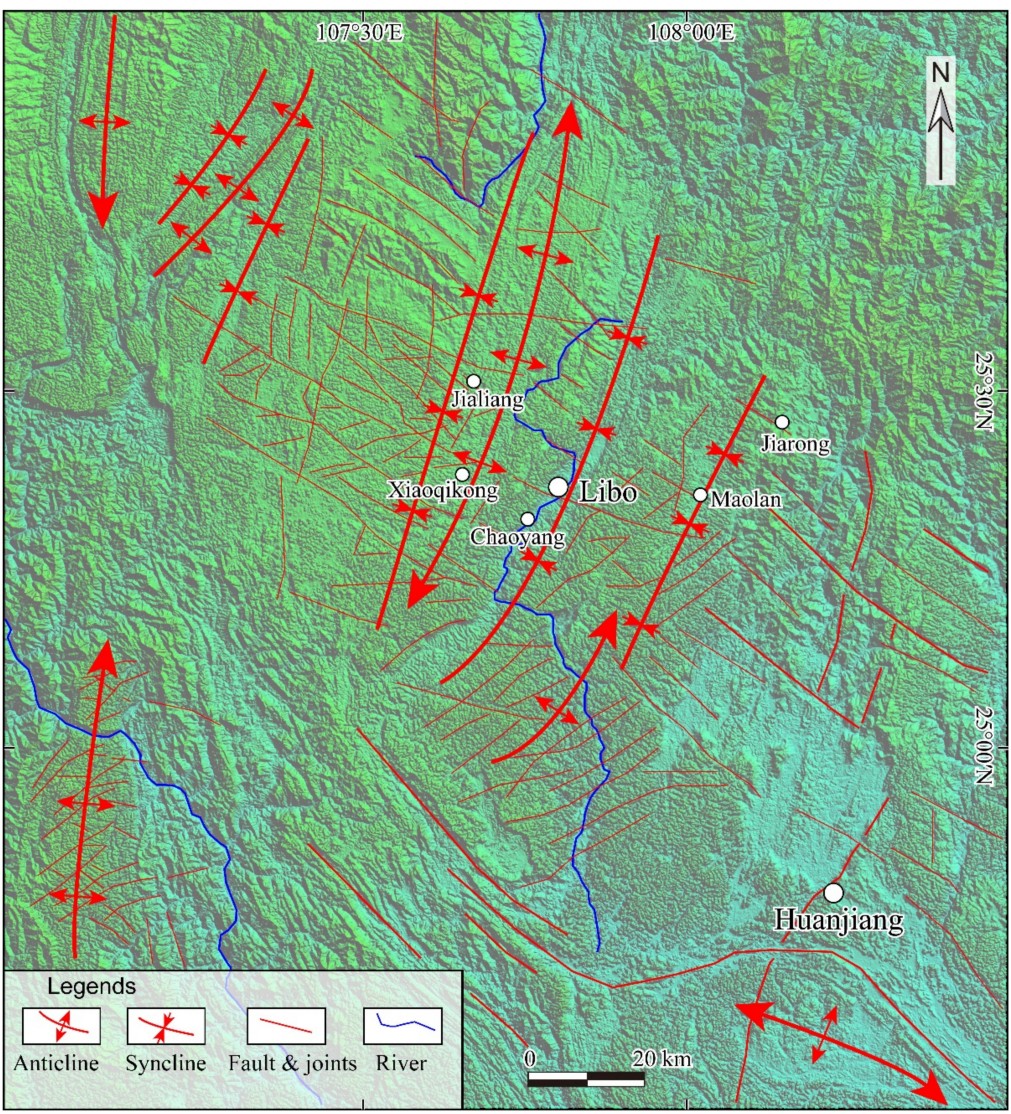

**Figure 2.** Structural features of the Libo area interpreted from DEM image.

Landsat 8 optical remote sensing image presented in Figure 3a displays the landscapes, vegetation and drainage system in Libo area. Figure 3a shows that Libo landscape is mainly vegetation-covered cone karst, distributed at interval in the EW direction. The most typical vegetation-covered cone karst landscapes are developed in the Maolan site (i.e., the area marked by a white dashed-line rectangle in Figure 3a and the corresponding sub-image

presented in Figure 3b). Besides, it also includes some carbonate hilly landforms covered by vegetation (mainly distribute in the northeastern region), as well as bare rocks and urban areas (marked by red dashed-line rectangles in Figure 3a). The Libo cone karst landscape has obviously controlled by the conjugate joints with distinct texture features. Due to the connection of the karst base, they appear as irregular light and dark spots on remote sensing images [38], looking like peanut shells as seen on the enlarged remote sensing image (Figure 3b).

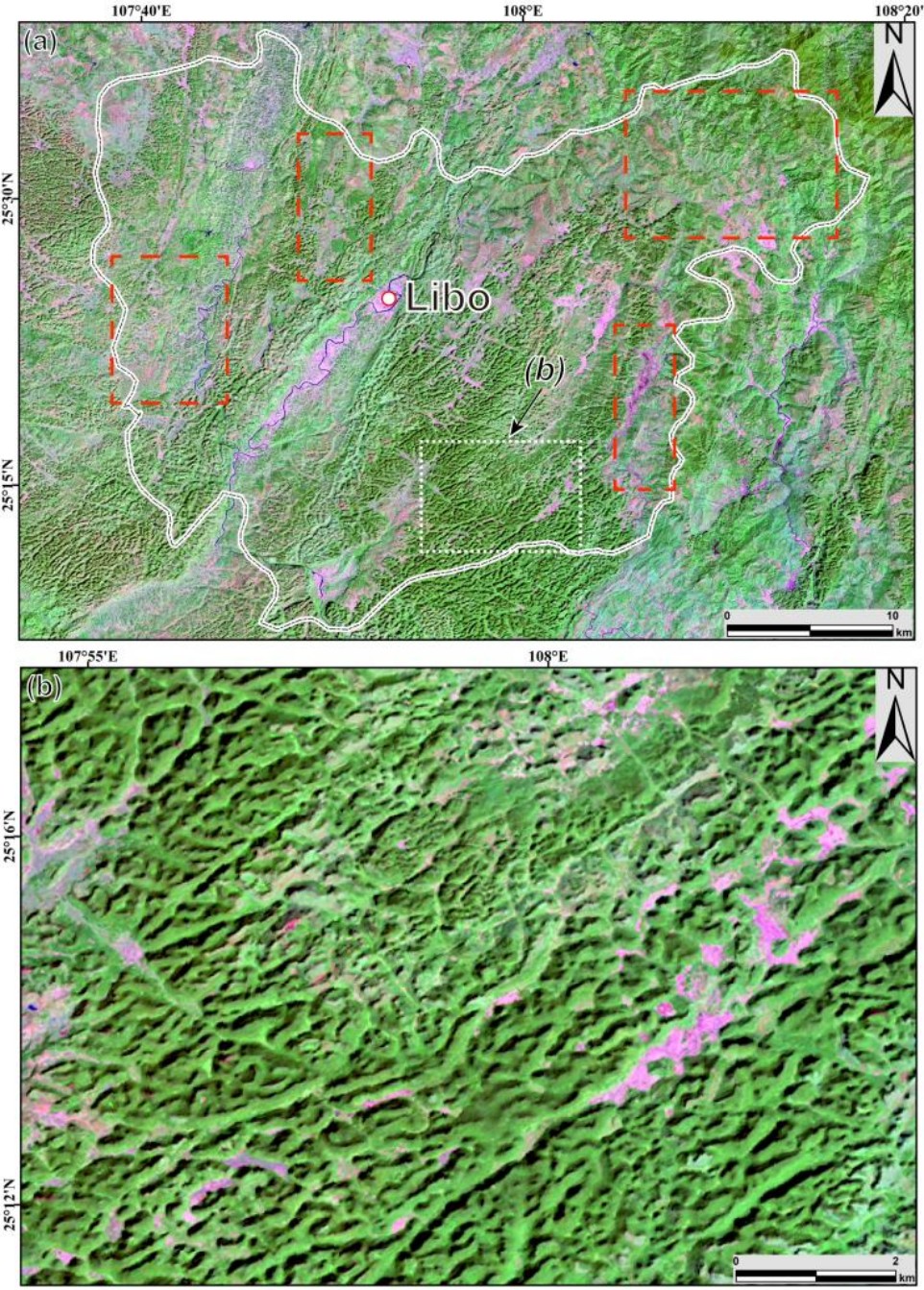

**Figure 3.** (**a**) Landsat 8 OLI image (RGB as 753) showing the geomorphological landscapes and vegetation of Libo area. (**b**) The sub-image corresponding to the area marked by the white dashed-line rectangle in (**a**) displaying typical peanut shell-like structure of the cone karst landscape in the study area.

## 3. Methodology

In this study, a segmentation method using both remote sensing and DEM data based on the DeepLab V3+ model was proposed to extract and map the cone karst landscape in Libo. The processing flow of the proposed method is shown in Figure 4.

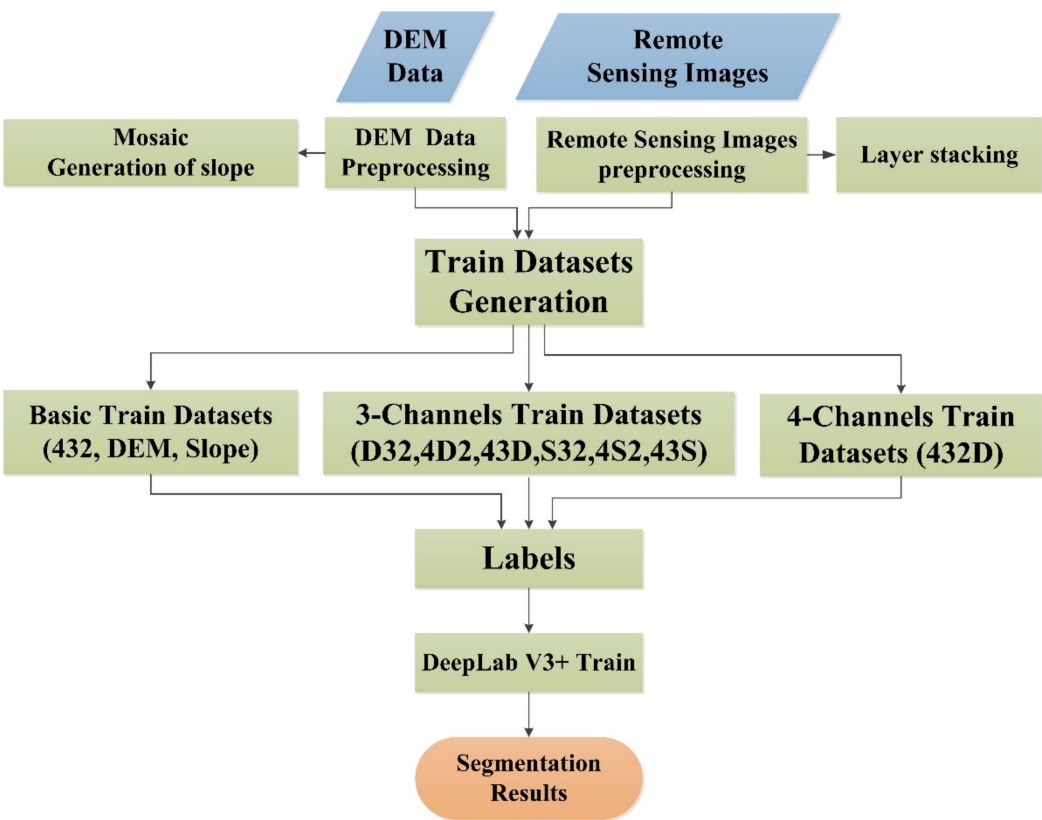

**Figure 4.** The processing flow of the proposed method to segment cone karst landscape based on the DeepLab V3+ model.

First, all remote sensing data and DEM data were preprocessed, including data import, layer stacking, cropping, stretch of DEM and the generation of slope. Secondly, we combined remote sensing data and elevation data from the same area by two ways: (1) replacing one of the RGB channels of remote sensing data with elevation or slope data to generate a new three-channel data; (2) generating a fused data with four channels of RGB and DEM. Thirdly, the label files of these training datasets were generated by the program of LabelMe in Linux system. Finally, an adjusted DeepLab V3+ model which can accept inputs of the four channels fused data with spectral, geometric as well as topographic information was used to achieve automatic extraction of cone karst landscape.

### 3.1. Training Datasets

The optical remote sensing images of Landsat 5, 7 (the bands arrangement is: R-4-Near Infrared band, G-3-Red band, B-2-Green band), and Landsat 8 (the bands arrangement is: R-5-Near Infrared band, G-4-Red band, B-3-Green band) were used as the training datasets. These images are Level 1T data products with spatial resolution of 30 m, which have undergone systematic radiometric calibration and geometric correction by United States Geological Survey (USGS). We collected a total of 86 scenes of Landsat image data with cloud cover less than 1%, acquired from July to September to generate the training datasets. During this period, the area is covered by vegetation, which is beneficial to distinguish cone karst landscape from urban area over Libo County, Guizhou Province and Huanjiang County (Figure 2), where having similar cone karst landscape with Libo [39] to generate

the training datasets. Additionally, the ASTER GDEM data with a spatial resolution of 30 m were also used to integrate with Landsat images.

### 3.2. Data Processing

#### 3.2.1. Data Preprocessing

As illustrated in Figure 4, the preprocessing of Landsat images is mainly layer stacking using the Environment for Visualizing Images (ENVI) software. For processing of the ASTER GDEM data, we firstly mosaic 25 original DEM data covering the whole area where the Landsat images were acquired, and then they were re-projected, so that they can be registered with the remote sensing images. Finally, the slope information was calculated based on the DEM data.

#### 3.2.2. Generation of Training Samples

#### Basic Training Samples

All the collected Landsat RGB images were divided to sub-images with a size of 500 × 500 pixels. We then selected sub-images presenting cone karst landscape (as positive samples) and non-karst landscape (as negative samples). Eventually, we obtained 898 sub-images which were further labeled for training the network (one such example is shown in Figure 5a). The sub-sets of the co-registered DEM data and calculated slope data corresponding to the Landsat sub-images were consequently extracted, as examples shown in in Figure 5b,c. Finally, three sets of basic training samples (each set has 898 training samples) were generated.

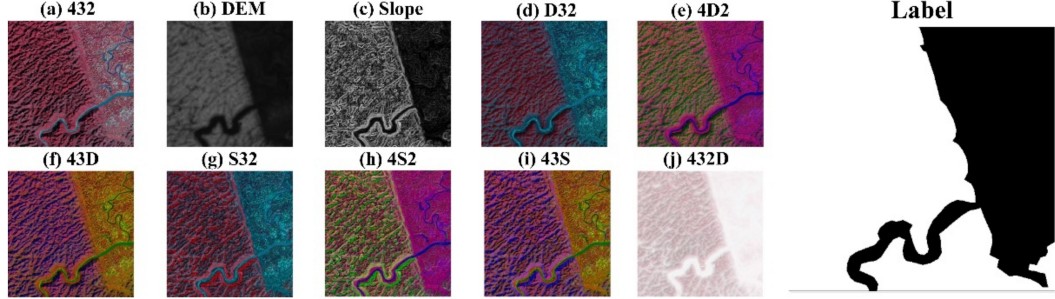

**Figure 5.** Examples of ten sets training samples generated in this study and the corresponding labeled result. (**a**) An example of optical remote sensing image training samples; (**b**) An example of DEM data training samples; (**c**) An example of slope data training samples; (**d**–**i**) Examples of three-channel fusion training samples; (**j**) An example of four-channels RGB-D training samples.

Due to special surface morphology and elevation information of cone karst landscape, we considered to fuse remote sensing images with the corresponding DEM data and DEM-originated slope data to generate additional training samples. The merged data contains both spectral, geometric and elevation or slope information of cone karst landscape. To test performances of different combinations of spectral and elevation data for classification of cone karst landscape, we made two types of fusion sample data, i.e., one is three-channel fusion data and another is four-channel RGB-DEM (abbreviated RGB-D hereafter) data.

#### Three-Channel Fusion Training Samples

By replacing one of the red, green, and blue channels of the Landsat remote sensing images with the co-registered DEM or slope data, we obtained six sets new three-channel training samples. For instance, by replacing the red channel (NIR band) information in the training sample shown in Figure 5a using the elevation data, a new training sample was generated, which is called D32, as shown in Figure 5d. Similarly, by replacing the green (red band) or blue (green band) channel information in the Landsat images using either the

elevation or slope data, more training samples were generated, as shown in Figure 5e–i, respectively.

Four-Channels RGB-D Training Samples

In this type of training samples, the DEM data was combined with the Landsat RGB images and consequently, we obtained four-channel training samples, i.e., red, green, and blue channels of remote sensing information and elevation information. The corresponding example is shown in Figure 5j.

Finally, a total of ten sets of training samples, which are expressed as: 432, DEM, Slope, D32, 4D2, 43D, S32, 4S2, 43S, 432D, were produced. Next, we outlined the edges of cone karst landscape of these images based on the program of LabelMe in Linux system. The right panel of Figure 5 shows the labelled result corresponding to the example of training sample. In the label image, the white part is cone karst landscape, and the black part is non-karst area.

### 3.3. DeepLab V3+ Model

DeepLab V3+ model is an advanced deep learning model for image semantic segmentation, with the goal of assigning semantic tags to each pixel in the input image. DeepLab V3+ applies Xception model for the semantic segmentation and the depth-wise separable convolution to both Atrous Spatial Pyramid Pooling and decoder modules, resulting in a faster and stronger encoder-decoder network [24].

The used DeepLab V3+ network architecture is shown in Figure 6, which has the same kernel function size, stride, activation function of the convolutional layer, pooling layer, and deconvolution layer as the one proposed by Chen et al. [24].

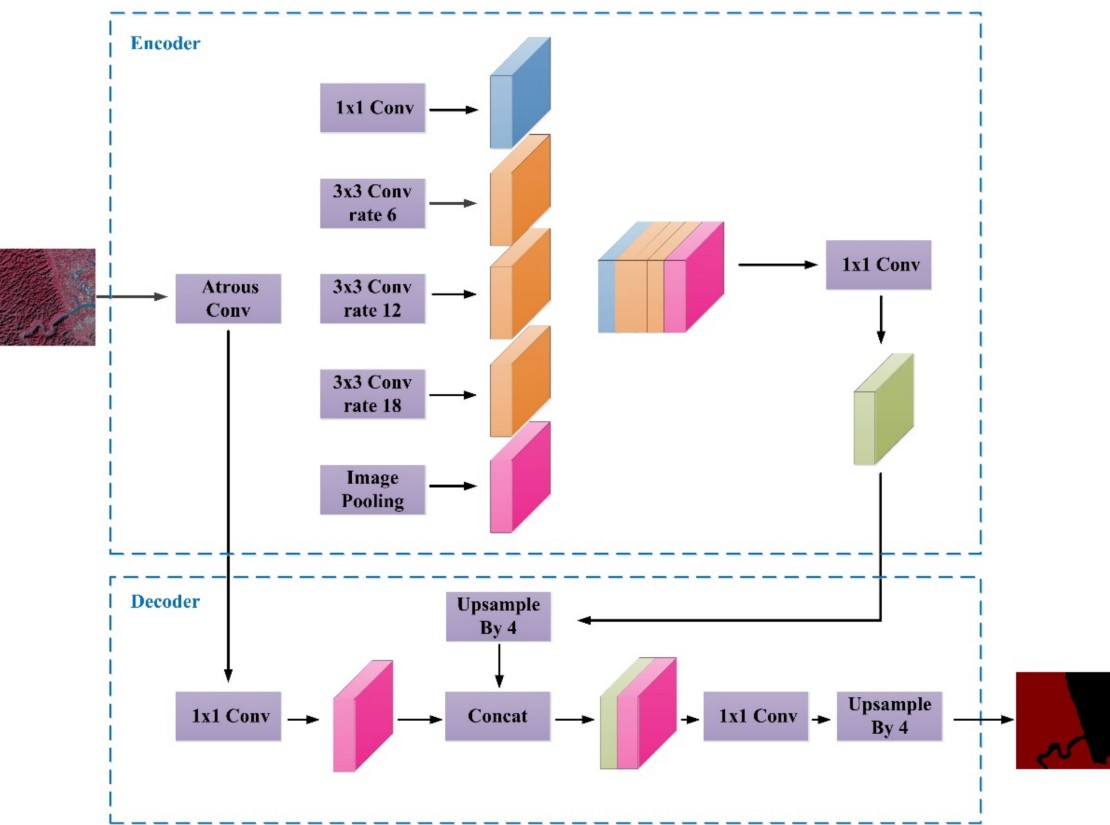

**Figure 6.** Network structure of the DeepLab V3+ model used in this study.

DeepLab V3+ model contains rich semantic information from the encoder module, while the detailed object boundaries are recovered by the simple yet effective decoder module, in order to achieve the purpose of improving the accuracy of segmentation. The difference is that we adjusted the input layer of DeepLab V3+ model so that the network could accept both the three-channel and four-channel samples shown in the left panel of Figure 5.

## 4. Segmentation Results of Cone Karst Landscape

### 4.1. Training Details of DeepLab V3+ Network

Training process of the DeepLab V3+ is based on the Tensorflow framework and the Python 3.7 in the Ubuntu environment. Technical specifications of hardware environment and software parameters of the training network are listed in Table 1. The numbers of training and test samples are 754 and 144, respectively.

**Table 1.** Technical specification of hardware and software used in this study for training of the DeepLab V3+ network.

|          |                |                 |
| -------- | -------------- | --------------- |
| **Hardware** | CPU            | Intel i7        |
|          | RAM            | 8GB             |
|          | Display Card   | NVIDIA TITAN XP |
|          | Display Memory | 12GB            |
|          | Hard Disk      | 2TB             |
| **Software** | Code Language  | Python 2.7      |
|          | Train Framework | Tensorflow      |
|          | Train Network  | DeepLab V3+     |
|          | Sample Label   | Labelme         |
|          | Train Times    | 40000           |

The datasets are modeled on the PASCAL VOC 2012 [32]. The original images are placed in the *JEPGImages* folder, while the label images are put in the *SegmentationClass* folder. In the Main folder, there are two text files: *train.txt* and *test.txt*, that respectively store the names of the training data and test data.

### 4.2. Segmentation Results

The MIoU is a general standard for evaluating the accuracy of deep neural network segmentation, which means the ratio of the intersection and union of the ground truth value set and the predicted value set. The MIoUs of the ten sets of samples (described Section 3.2) used for training the DeepLab V3+ network are listed in Table 2.

**Table 2.** MIoUs of different samples used for training the DeepLab V3+ network.

| **Sets** | **MIoU** |
| -------- | -------- |
| 432      | 92.10%   |
| DEM      | 88.78%   |
| Slope    | 59.15%   |
| D32      | 91.92%   |
| 4D2      | 91.72%   |
| 43D      | 93.96%   |
| S32      | 91.62%   |
| 4S2      | 92.86%   |
| 43S      | 92.39%   |
| 432D     | 95.53%   |

Table 2 shows that the slope data used as training data has the lowest MIoU value, followed by only using the DEM data. This suggests that only elevation or slope data, though which is a key factor of cone karst landscape, is not sufficient to classify them without considering cone geometry. Among the seven sets of three-channel data, the training accuracy of 43D (i.e., combination of Landsat NIR, Red bands and DEM data) is the highest. A plausible reason is that the wavelength of green band is shorter than that of near infrared band and red band in remote sensing image, so it is affected by the atmosphere more. By replacing the green band of the Landsat data by the DEM data, the influence of atmosphere on remote sensing image is reduced while the elevation information is also included, therefore, the segmentation accuracy is higher than the results using other three-channel training samples. The MIoU of the four-channel training data (i.e., 432D) is the highest, which can reach 95.53%. This result indicates that the fusion of remote sensing and DEM data is effective to improve the recognition accuracy of cone karst landscape.

Two typical regions were selected in the test dataset to compare the segmentation results based on different training samples, as presented in Figure 7a,b, respectively. The left panel in the two figures presents ground truth images, and the right part is the segmentation results using ten different training datasets. The red and black in the segmentation images are the cone karst landscape and non-karst landscape classified by DeepLab V3+ model, respectively.

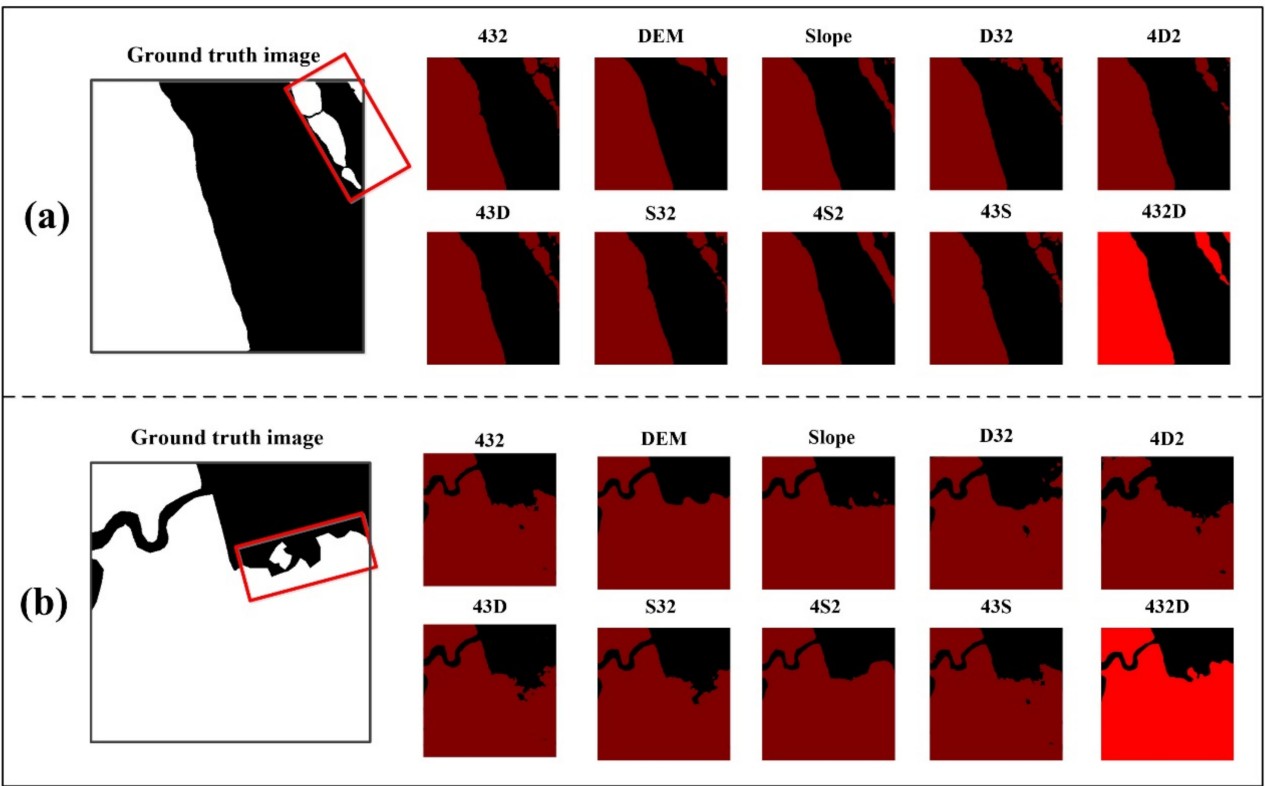

**Figure 7.** Demonstration of segmentation results of (**a**) A case presenting discrete distribution of cone karst landscape using ten different training datasets and (**b**) A case presenting variable boundary of cone karst using ten different training datasets.

Although most of the cone karst landscapes are connected by bases, there are still a few scattered cone peaks. The case in Figure 7a shows classification result for discrete distribution area of cone karst landscape (as marked by the red rectangle) based on the proposed method. Comparing the segmentation results using the different ten sets of training samples, 432, 43D, S32, 43S and 432D have similar recognition effects and perform well; DEM, Slope, 4D2 and 4S2 have missing some areas; D32 has misclassification phenomenon.

Figure 7b presents an example to demonstrate segmentation result for the boundary recognition of the cone karst landscape (as the area marked by the red rectangle). 432D performs the best; 432, 4D2, 43D and 43S have the similar recognition effects and perform well; DEM, Slope, S32 and 4S2 have missing some points; D32 has misclassification phenomenon.

### 4.3. Comparison of the Segmentation Results by SVM and DeepLab V3+

SVM is a general machine learning method based on the Statistical Learning Theory (SLT) proposed by Vapnik in 1963 [40]. The aim of the SVM for classification is to determine a hyper plane that optimally separates two classes. An optimum hyper plane is determined using train data sets and its generalization ability is verified using test data sets [41]. By realizing the idea of structural risk minimization (SRM), SVM overcame the problems of overfitting and nonlinear in traditional machine learning methods. Additionally, it can effectively deal with the high-dimensional data problem with the help of kernel function, generally aggregated into four groups: linear, polynomial, radial basis function (RBF) and sigmoid kernels [41], so SVM has strong nonlinear disposition ability and generalization performance [42,43]. The SVM method used in this study was implemented by ENVI, the kernel function was chosen RBF.

Figure 8a,b show the segmentation results by using SVM and DeepLab V3+, respectively. The green area displays the cone karst landscape. The marked red rectangles highlight those distinct different segmentation results between these two methods. In fact, the segmentation result by using the DeepLab V3+ network is quite different from that achieved using the SVM, though SVM is also a good two-class classifier. The results show that the two methods are quite different in recognizing carbonate hilly landscape which has the same vegetation cover with cone karst landscape (as the areas marked by the red rectangles in Figure 8). But the relief contrast of the carbonate hilly landscape is not as obvious as the cone karst landscape, as the area marked by the red rectangle in Figure 3a. As can be seen from Figure 8, the DeepLab V3+ model can successfully distinguish the cone karst landscape from the carbonate hilly landscape, while the SVM method misclassifies the carbonate hilly landscape into cone karst landscape.

A field investigation was carried out in early November 2020 to verify the segmentation results based on satellite remote sensing data and numbers of field photographs. The field investigation photograph shown in Figure 9a was acquired over the Dongduo village belonging to the Maolan site, corresponding to the location A marked in Figure 8, which is classified cone karst by both SVM and DeepLab V3+ network. In contrast, the photo in Figure 9b was acquired over Jiarong town (location B marked in Figure 8) shows a carbonate hilly landscape. The segmentation result using DeepLab V3+ correctly classify it to non-karst landscape whereas the SVM misclassifies the area. The comparison of segmentation results using different machine learning methods and the verification by field investigations further suggest that the DL-based method is powerful for classifying cone karst from remote sensing data.

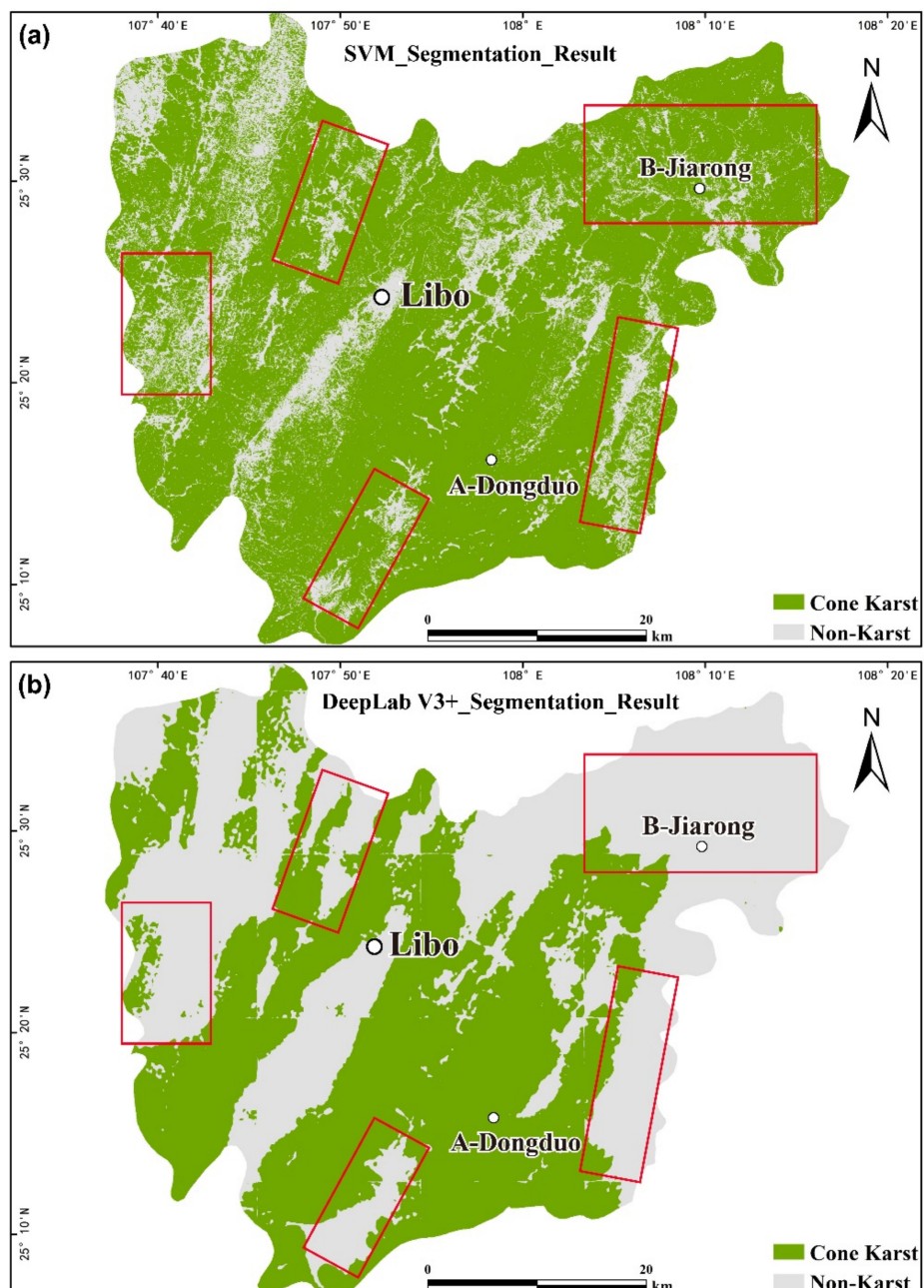

**Figure 8.** Segmentation results of cone karst in Libo using (**a**) SVM model and (**b**) DeepLab V3+ model.

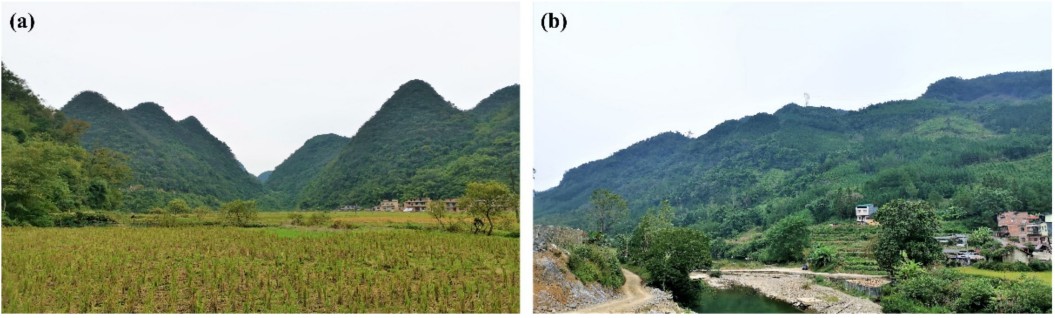

**Figure 9.** Field photographs showing the typical geomorphic landscapes in the Libo. (**a**) Cone karst landscape observed at Dongduo village, Maolan area; (**b**) Carbonate hilly landscape observed near Jiarong town.

## 5. Discussion

How to extract automatically and effectively the geomorphic landscapes with the irregular geometric features from remote sensing data is a tough problem. Cone karst landscape is an important part of UNESCO Natural Heritage South China Karst. Its mapping from space, particularly in terms of geomorphic information and spatial distribution is of great importance for scientific conservation, management and eco-tourism development for heritage site. Satellite remote sensing images and field investigations show that geomorphic landscapes in Libo are quite complicated (Figure 9). In addition to the dominant cone karst landscape, the low mountains and carbonate hilly landforms as well as town and farmland also exist in Libo heritage site. Particularly, the cone karst landscape and carbonate hilly landscape are both covered with vegetation and therefore their spectral features are similar in optical remote sensing images. This increases difficulties of visual inspection of cone karst landscape and also reduces accuracy of classification using only remote sensing images based on a certain machine learning method. However, the relief contrasts of these two landscapes are different. The relief contrast of cone karst landscape is between 100 m to 250 m, while the relief contrast of carbonate hilly landform is mostly between 50 m to 100 m as we measured from the images.

Therefore, to better classify the cone karst and non-karst landscape in Libo, we combine optical remote sensing images containing the spectral and geometric information with the DEM data including the topographic information to generate a four-channel training samples for DeepLab V3+ network. However, DeepLab V3+ network can only accept either one-channel or three-channel data as input, to accomplish four-channel data used as training samples, we adjusted the input module of DeepLab V3+ network. MIoU of the segmentation result achieved using the four-channel training samples (i.e., 432D data) is of 95.53%, which is higher than that using remote sensing image data (i.e., 432 with MIoU of 92.10%) alone. Even replacing the green band in general Landsat RGB data by the DEM data, the segmentation can also achieve a higher accuracy of 93.96% than only RGB data used as training samples. These results indicate the additional DEM information used for training the DeepLab V3+ is particularly effective to recognize the cone karst landscape in Libo.

Compared with the SVM method, the segmentation effect of DeepLab V3+ model is more accurate for the recognition of the cone karst landscape and the carbonate hilly landform with the same input data (Figure 8).

Additionally, the distribution of the cone karst landscape recognized by DeepLab V3+ model is consistent with the position of anticlines presented in Figure 2. But the syncline areas are easy to converge with the surface and ground water because of the low terrain, which aggravates the erosion of the syncline axis. So that the cone karst landscape cannot develop and preserve well. However, the faults and tensile joints or fractures are well developed along the anticline structures, which are conducive to the infiltration and erosion of surface water into the depth, thereby promoting the development of cone karst landscape.

The optimal accuracy of DeepLab V3+ model can reach 95%, which is much higher than 80% of the traditional interpretation method [12], which suggests that deep neural network model can be applied to extract automatically the geomorphic information and spatial distribution of cone karst landscape effectively.

## 6. Conclusions

In this study, we newly developed a method based on deep neural network model to extract automatically and effectively landscape information of cone karst using both optical satellite remote sensing data and DEM data. The main conclusions are as follows:

1.  An improved DeepLab V3+ network has been proposed to carry out semantic segmentation using multi-source remote sensing images. A set of machine learning samples of cone karst landscape and a set of four-channel data were further obtained from the fusion of optical remote sensing image and DEM data. The input module

of DeepLab V3+ model was adjusted so that the network can accept multi-channels (four channels or more) data.

2. It is found that the segmentation results using different combinations of four-channel data (i.e., 432D) has the highest accuracy of 95.53% in terms of MIoU. The second highest accuracy is that three-channel training data of 43D because the green band of Landsat data was replaced by the DEM data, which can largely reduce the atmospheric influence on Landsat image. Thus, the segmentation accuracy is higher than the results using other three-channel training samples.

3. Compared with the SVM method, the major advantage of segmentation results from the DeepLab V3+ model is that the deep neural network structure can effectively distinguish the cone karst and carbonate hilly landscapes through multiple deep convolution layers and pooling layers as verified by the field observation and UAV investigations.

In general, this study demonstrates that the deep learning-based pixel-level image segmentation method of automatic recognition of cone karst landscape can effectively solve the problems of low efficiency and low precision in recognition and classification of irregular and complex geological landforms using remote sensing image, which provides a powerful method for mapping the worldwide cone karst landforms.

**Author Contributions:** Conceptualization, H.F. and B.F.; methodology, H.F.; software, H.F.; validation, H.F. and B.F.; formal analysis, H.F.; investigation, H.F., B.F. and P.S.; resources, B.F.; data curation, H.F. and B.F.; writing—original draft preparation, H.F.; writing—review and editing, H.F. and B.F.; visualization, H.F. and B.F.; supervision, B.F.; project administration, B.F.; funding acquisition, B.F. All authors have read and agreed to the published version of the manuscript.

**Funding:** This research was supported by the Second Tibetan Plateau Scientific Expedition and Research Program (STEP) (2019QZKK0901), the Strategic Priority Research Program of Chinese Academy of Sciences (XDA 20070202).

**Institutional Review Board Statement:** Not applicable.

**Informed Consent Statement:** Not applicable.

**Data Availability Statement:** Data sharing is not applicable to this article.

**Acknowledgments:** We really appreciate two anonymous reviewers for their constructive suggestions and comments that help us to improve the quality of this paper. Landsat remote sensing data and GDEM data used in this paper were downloaded from China Geospatial Data Cloud website: http://www.gscloud.cn.

**Conflicts of Interest:** The authors declare no conflict of interest.

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
