# Peer review of "An Improved Segmentation Method for Automatic Mapping of Cone Karst from Remote Sensing Data Based on DeepLab V3+ Model"

_remotesensing, doi:10.3390/rs13030441_

Round 1

Reviewer 1 Report

Dear Editor and Authors, I had the privilege of reading and evaluating the paper “An improved segmentation method for automatic mapping of cone karst from remote sensing data based on DeepLab V3 + model”. The paper presents important information in the area of remote sensing, but there is still a need to improve on some points that I mention below:

  1. Why was this type of technique / sensory analysis performed? I did not find any scientific justification in the introduction, I believe it is important to mention with the authors' own words a paragraph addressing the importance of this study for this region and for others worldwide;
  2. Figures 1, 2, 3, and 4 can be joined together in a single figure. The authors know their area of study, but it is very difficult to know their region at the globe level, therefore, I suggest that the area of study can include, the Country, State, city and other information that allows the reader from other regions of the world globe locate;
  3. The authors comment that they used Landsat 5, 7 and 8. Were the images corrected? How were they corrected? It is known that the Landsat 5 images present some problems of projection and overlap with vector data. How were they corrected? Has corrected NASA images been used? If so, authors should mention this in the text. In addition to the necessary overlap and corrections, was only the raw image used? Or was the radiometric calibration and reflectance calculated? What equation? Was it used images already corrected and calculated by NASA?
  4. Join Figures 8 and 9 into a single Figure, perhaps one below the other (a and b). There are many figures in the work, I believe that the authors can endeavor to reduce this number.

Reviewer 2 Report

The main objective of this paper is the improvement of landscape classification accuracy and efficiency. It proposes an automatic method to extract spatial distribution of cone karst landscapes by combining satellite remote sensing images and digital elevation model data based on DeepLab V3+, which can be considered as a methodological originality. Concerning presentational issues, the abstract, keywords, and introduction are o.k. The importance of study is highlighted, the purpose of the work is presented, and key publications are cited. Study area is sufficiently described in section 2, but authors should make a distinction between m and m a.s.l. A very detailed explanation of methodology is presented in section 3. The obtained results are presented in section 4. In this section, to highlight the advantage of using DeepLab V3+ network, segmentation based on SVM was also conducted. The problem is that SVM is not explained. There is no explanation of method/model, reference, or even abbreviation SVM. SVM is not also mentioned in the introduction. Discussion seems o.k., but conclusions should be more concentrated on the main findings, instead of describing again the method and results.
